# Evaluating the Pattern of Relationships of Speech and Language Deficits with Executive Functions, Attention Deficit/Hyperactivity Disorder (ADHD), and Facets of Giftedness in Greek Preschool Children. A Preliminary Analysis

**DOI:** 10.3390/bs15020136

**Published:** 2025-01-26

**Authors:** Maria Sofologi, Georgia Chatzikyriakou, Chrysoula Patsili, Marina Chatzikyriakou, Aphrodite Papantoniou, Magda Dinou, Eleni Rachanioti, Dimitris Sarris, Harilaos Zaragas, Georgios Kougioumtzis, Dimitra V. Katsarou, Despina Moraitou, Georgia Papantoniou

**Affiliations:** 1Laboratory of Psychology, Department of Early Childhood Education, School of Education, University of Ioannina, 45110 Ioannina, Greecemarinacha@hotmail.com (M.C.); afropapanto@gmail.com (A.P.); mdinou@uoi.gr (M.D.); e.rachanioti@uoi.gr (E.R.); gpapanto@uoi.gr (G.P.); 2Institute of Humanities and Social Sciences, University Research Centre of Ioannina (U.R.C.I.), 45110 Ioannina, Greece; 3Department of Psychology, School of Health Sciences, Neapolis University Pafos, Paphos 8042, Cyprus; georgetype@gmail.com (G.K.); d.katsarou@aegean.gr (D.V.K.); 4Pedagogy and Teaching Methodology Laboratory, Department of Early Childhood Education, School of Education, University of Ioannina, 45110 Ioannina, Greece; sarrisdem@gmail.com (D.S.); hzaragas@gmail.com (H.Z.); 5Department of Turkish Studies, National and Kapodistrian University, 11528 Athens, Greece; 6Department of Preschool Education Sciences and Educational Design, Faculty of Humanities, University of the Aegean, 85131 Mytilene, Greece; 7Laboratory of Psychology, Section of Experimental and Cognitive Psychology, School of Psychology, Aristotle University of Thessaloniki, 54124 Thessaloniki, Greece; demorait@psy.auth.gr; 8Laboratory of Neurodegenerative Diseases, Center for Interdisciplinary Research and Innovation (CIRI-AUTH) Balkan Center, Buildings A & B, Aristotle University of Thessaloniki, 57001 Thessaloniki, Greece

**Keywords:** speech and language deficits, teachers’ evaluations, executive functions, preschool children, giftedness

## Abstract

Speech and language deficits often occur in preschool children, and empirical studies have indicated an association between language impairments and challenges in different cognitive domains. The primary aim of the current study was to evaluate the associations between speech and language deficits, executive function (EF) impairments, Attention Deficit/Hyperactivity Disorder (ADHD), and aspects of giftedness in Greek preschoolers based on assessments from their teachers. Investigating the associations between aspects of EFs, ADHD, and giftedness was another objective of the current study. Finally, we examined on a sample of Greek preschool children the convergent validity of the LAMP screening test in relation to the following questionnaires: the Childhood Executive Functioning Inventory (CHEXI), the ADHD-IV Questionnaire, the Gifted Rating Scales-Preschool/Kindergarten Form (GRS-P), and the Scales for Rating the Behavioral Characteristics of Superior Students (SRBCSS). For the purpose of the present study, 20 kindergarten teachers and 71 Greek preschoolers (41 boys and 30 girls) were included in the sample. Data analysis revealed that according to teachers’ estimations, speech and language deficits are positively associated to a statistically significant degree with ADHD and with deficits in working memory (WM) and inhibition. On the other hand, aspects of preschool-aged creativity and giftedness were significantly correlated negatively with speech and language deficits. Additionally, the findings demonstrated a negative correlation between aspects of giftedness and ADHD symptoms as well as poor achievement on working memory (WM) and inhibition assessment tests. Furthermore, there was no association between hyperactivity/inhibition deficit and creativity, which is an aspect of giftedness. The moderate positive associations of the LAMP screening test with the psychometric tools of measurement of ADHD and executive function (EF) deficits, and the negative associations with the scales of giftedness showed the good convergent and distinct validity of the LAMP assessment test.

## 1. Introduction

Language is a fundamental component of all social and intellectual experiences since it is used for knowledge analysis, organization, and discussion. Children’s learning, social, and emotional development depend heavily on their capacity to use language as a social and educational tool to enhance social connections and academic achievement ([51]). The first year of life is the foundation for language exchanges, as demonstrated by a series of studies ([73]; [13]). When we examine the early beginnings of language in human life, we can observe the earliest indications that children are actively learning the language they are exposed to and can comprehend the meaning of particular words when they are 9 or 10 months old ([31]; [32]). Language development is crucial for both general cognitive and social advancement as well as for the early phases of developmental pathways. The complicated process of language acquisition necessitates a variety of linguistic skills and cognitive functions, including EFs. The development of language proficiency varies significantly between early and later in life. Some children begin language development early and pick it up quickly, while others who start later do reach normal language levels by the middle to late preschool years ([13]; [51]). Specifically, the rate and patterns of language development in children differ. Some children may begin talking later than others, but their speech and language development continues to advance rapidly, and there seems to be no reason for additional concern. Other children may begin early and acquire speech and language quickly ([31]; [32]). However, the development of some children is challenging since their speech and language abilities are delayed, necessitating more care and attention ([56]). Language acquisition can be difficult for some children, which can affect how well they do in school and how they interact with others. Evidence suggests that one out of ten children in all age groups have language and communication problems ([56]). By the age of five, for instance, approximately 50% percent of late-talking children seem to have caught up with their peers. Additionally, 50% to 90% percent of these children will continue to have language impairment throughout childhood if the language delay is still present at kindergarten age, and many of them may experience concomitant behavioral, mental, or academic challenges in the future ([13]; [73]). Moreover, 1% of these children have the most severe and complex speech and language deficits (SLDs), and the rates are higher in socially deprived communities, where up to 50% of children have speech and language abilities that are below average for their age. Preschoolers experience challenges at a high rate, as might be predicted, but by the time they start school, the rates appear to decline ([13]; [31]; [32]; [51]; [73]). However, these challenges vary in intensity and are likely to be resolved in the early school years. On the other hand, peech and language deficits during childhood tend to remain throughout development, which has lifetime repercussions for academic, social–emotional, and behavioral functioning ([13]; [31]).

The etiology of speech and language deficits involve an amalgamation of environmental, genetic, and epigenetic factors in which academic skills fall well short of predicted levels despite age, IQ, and appropriate education ([56]). One or more components of academic functioning, such as reading, writing, and mathematical ability, are affected by an impaired learning profile ([32]; [56]).

Speech and language deficits, in the form of poor spoken language and comprehension abilities, and Attention-Deficit/Hyperactivity Disorder (ADHD) are two relatively common developmental disorders that have important consequences in the lives of the affected individuals. Attention Deficit/Hyperactivity Disorder (ADHD) and speech and language deficits are two prevalent conditions that can impact a child’s language skills. Specifically, speech and language deficits often occur in preschool children, and research has shown interdependent associations of language impairments with difficulties in cognitive areas such as executive functions (EFs) and working memory (WM) ([31]; [32]). There is consistent evidence that a considerable percentage of children with ADHD have delayed language development, which impacts their academic ability in reading and their mathematical ability as well as early speech onset ([14]; [40]; [32]; [56]). According to current studies, 40% of children with ADHD have difficulties with phonological processing ([11]), and about 30% of them show notable delays in their reading ability ([14]; [40]; [56]). Children with the combined form of ADHD face these difficulties more acutely ([11]). Low vocabulary, difficulties in acquiring new words, a preference for simpler sentence structures, omitting sentence components, avoiding complex expressions, and a slow speaking rate all contribute to the educational challenges that children with ADHD face. Such challenges are persistent, reflecting the intricate obstacles these children navigate, exacerbated by core symptoms of ADHD, including distractibility and executive function (EF) deficits ([11]; [42]). 

As stated by [56] ([56]), EFs are required for the regulation of more automatic processes (thoughts, behavior, and emotion) in support of goal-directed activities. EFs are described as the top-down control of cognitive processes for goal achievement. Miyake’s model ([56]), a well-known conception of EF, suggests that EF is a unified construct with three distinct components: mental set shifting, inhibition, and information updating/monitoring of WM representations. Research findings show that children with language disorders and ADHD characteristics frequently have co-morbid deficiencies, including impairments in EFs ([14]; [40]).

Preschool children with speech and language deficits and behavioral problems display difficulties in measures assessing WM, cognitive flexibility, and inhibition ([11]; [42]). In specific, [91] ([91]) in their research revealed that EFs of preschool children with speech and language deficits were found to be significantly poorer than those of their classmates, according to parents and teachers’ evaluations. Additionally, impairments in inhibition, emotional regulation, and organization were disclosed in children aged 4–5 years old according to their performance in different behavioral and WM assessments tests ([87]). This association or co-occurrance between EF deficits and learning challenges in neurodevelopmental disorders, like ADHD and speech and language deficits, is supported by research findings ([21]; [58]). EF impairments, such as WM, cognitive flexibility, attention, and inhibitory control, might arise in preschool children and school age children with comorbid speech and language deficit and ADHD while in classroom learning ([4]; [54]; [80]).

According to research studies in typically developing preschool children, important advancements in EF are made during this time ([71]). This significant improvement in EFs aligns with evidence of structural modifications in the prefrontal cortex during early childhood, a brain region known to play a significant role in EF ([59]). Furthermore, it has been proposed that EF develops in a hierarchical manner, with attention acting as the cognitive foundation: during the first three years of life, simpler EF components (like behavioral inhibition) develop and are then integrated into more complicated EF processes (like planning) ([31]).

Literature analysis indicates that preschoolers with speech and language deficits may struggle with WM, which is essential for updating EFs, as well as tasks assessing inhibitory control and cognitive flexibility ([85]). Additionally, according to teachers’ evaluations, preschool children with language deficits exhibit higher levels of hyperactivity than their peers ([86]). There appears to be a link between language deficits and hyperactivity. [24] ([24]), has suggested that language processing difficulties may be the fundamental cause of many of the externally evident diagnostic features of ADHD. Furthermore, research has highlighted that speech and language deficits are a frequent comorbidity found in children with ADHD ([1]; [50]).

More recently, in the sphere of education, the research community have placed emphasis on the fact that there is growing research attention on the relation between giftedness and ADHD characteristics in preschool children. Some children exhibit both giftedness and ADHD traits. Research analysis has revealed that children with both giftedness and ADHD characteristics received little attention in worldwide educational research and educational programming until the last ten years. There is recent evidence that dual diagnosis is valid. Empirical evidence shows that gifted children with ADHD characteristics have a pattern of behavioral, mental, and cognitive traits that are in line with the ADHD diagnosis recorded in children with average IQs ([1]). The prevalence of ADHD in gifted populations is also consistent with ADHD in the general population ([63]). In parallel, [63] ([63]) define high-privileged or gifted children as those who, in comparison to many of their peers, can complete the curriculum at a significantly faster pace and at a significantly higher level of abstraction and difficulty due to their developed cognitive and creative skills, traits, motivations, and preferences. Research on gifted children employs a comprehensive definition of giftedness that incorporates a range of traits, abilities, and talents that can be proven in many ways ([68]). On the other hand, researchers are focused also on “twice exceptional” gifted children, a concept that is characterized by a potential for high performance in skills or creativity in one or more areas of activity coupled with the presence of a mental health or neurodevelopmental disorder such as ADHD ([66]). The relevant literature argues that the term gifted with speech and language deficits connotes “twice exceptional” children (2E) ([5]). These children exhibit giftedness in one or more academic field, as well as speech and language deficits ([27]). Consequently, these children’s strengths are attributed to giftedness, whereas their weaknesses are attributed to speech and language deficits. Despite their generally great potential, gifted children may not always meet expectations. These students have been officially classified as gifted in addition to having co-occurring diagnoses of special education needs. The intensity, sensitivity, impatience, and high motor activity of gifted students might readily be confused with an ADHD profile ([92]). Within this framework, some of the ADHD criteria are behaviors that are often referred to in the characteristics of gifted children. Among the characteristics of giftedness involved in ADHD is the difficulty of adapting to new conditions, impulsiveness, and fast speech ([45]). [38] ([38]) conducted the first empirical study to illustrate the potential misdiagnosis of ADHD and giftedness. [38] ([38]) hypothesized that a number of issues common to gifted children would complicate ADHD misdiagnosis because of an overlap in symptoms or behaviors characteristic of giftedness and those characteristics of ADHD, including high activity levels, difficulty paying attention, and impulsiveness. It has been suggested that asynchronous development ([78]), classroom boredom ([88]), or overexcitabilities are among the causes for these behaviors among gifted children ([30]). Research analysis shows that when ADHD characteristics coexist with giftedness, ADHD may impair EFs including WM, time duration processing, and auditory verbal memory ([30]). Impairment of EFs may indicate significant challenges in reaching a level of academic performance consistent with the child’s intellectual capacity ([38]; [45]).

### The Present Study

Gifted children with ADHD characteristics tend to demonstrate deficits in WM and EFs compared to gifted children without ADHD ([3]; [17]). Children who were diagnosed with both ADHD and giftedness received little consideration in educational programs and research until the last decade. Individuals who have two exceptionalities, both giftedness and a learning or language disorder—have been neglected in education and research endeavors in the past ([28]). Dually diagnosed children still experienced academic underachievement, difficulty organizing, and difficulty maintaining attention. Interestingly, a research overview on the reciprocal relationships between giftedness and ADHD characteristics would be lacking empirical data. Designing successful educational policies and programs for children with gifted and ADHD characteristics requires a closer research look at and analysis of these interaction possibilities in order to illuminate this research gap. Identifying the links or the pattern of interconnections between gifted students’ Attention-Deficit/Hyperactivity Disorder (ADHD) and speech and language underachievement is a key factor that has not received much empirical research according to research overview ([3]). [3] ([3]) noted that gifted preschool and school children with ADHD characteristics show a pattern of cognitive, psychiatric, and behavioral characteristics consistent with the diagnosis of ADHD documented in children of average IQ. Additionally, the research findings of [20] ([20]) showed that 10 of the 15 gifted children in their clinical trial fulfilled the DSM-IV criteria for ADHD, which validated dual diagnosis. Additionally, ADHD, learning and speech disabilities, and EF issues have been also identified in a small number of gifted children ([90]). The prevalence of giftedness with learning deficits is thought to be between 2% and 15% of the overall population from preschool age up to adolescents ([89]). Furthermore, children with a giftedness and ADHD profile are notable for being characterized with high levels of creativity ([29]). It appears that creative thinking and ADHD are associated. Specifically, a series of studies have indicated that children with ADHD have a great capacity for creativity ([52]). Creativity is the process of using information that appears unrelated and unimportant to solve issues. This implies that a broad focus of attention, which is frequently defined as distractibility and is known to occur in people with ADHD, could be beneficial for original thought and insight ([75]). Additionally, there seems to be a negative correlation between creativity and EFs. Research has shown that creativity increases with decreasing WM capacity ([29]). Moreover, the findings of [19] ([19]) revealed that children with high levels of creativity also have impairments in inhibition.

Under the auspices of the above empirical findings, the main purpose of the present study is to shed light on the pattern of relationships between speech and language deficits with EF deficits and Attention Deficit/Hyperactivity Disorder (ADHD) in preschool children. In line with this aim, we evaluated whether deficits in EF and ADHD measurements are related to language deficits according to the LAMP test. Additionally, the present research aimed to investigate the relationship of EFs with ADHD, and the relationship between these two parameters with different facets of giftedness in Greek preschool children according to their teachers’ evaluations. Finally, another aim of the present study was to evaluate, in a sample of Greek preschool students, the convergent validity of the Linguistic Assessment for Mapped Provision test (LAMP) ([60]) in relation to the measurement of EFs, ADHD, and different facets of giftedness. In specific, we hypothesized that impaired language skills are positively correlated with deficits in EFs (WM and Inhibition) (Hypothesis 1a). Furthermore, speech and language deficits are positively correlated with ADHD characteristics (Hypothesis 1b). Speech and language deficits are negatively correlated with facets of giftedness (Hypothesis 1c). In the context of our second objective, we hypothesized that different facets of giftedness are negatively correlated with ADHD (Hypothesis 2a). Also, different facets of giftedness are negatively associated with EF impairments (Hypothesis 2b). We also presumed that the ADHD symptoms are positively correlated with EFs (Hypothesis 2c). Finally, for the structural and convergent validity of the LAMP test with the Greek version of the Childhood Executive Functioning Inventory (CHEXI) ([82]) and the ADHD-IV Inventory ([26]), it is expected to achieve a positive correlation between these measurements (Hypothesis 3a). The Linguistic Assessment for Mapped Provision test (LAMP) ([60]) is expected to correlate negatively with all three sub-scales of the Gifted Rating Scale-Preschool Form Kindergarten Form (GRS-P) ([62]), and five scales of the SRBCSS ([69]) reflecting the level of discriminant validity between the assessment tests (Hypothesis 3b). For the purpose of the present study, facets of giftedness were evaluated with the first three scales of the Gifted Rating Scales-Preschool/Kindergarten Form (GRS-P) ([62]) (Intellectual ability Scale, Academic ability Scale, and Creativity Scale) and five scales from the Scales for Rating the Behavioral Characteristics of Superior Students (SRBCSS) ([69]) (Learning, Creativity, Communication (precision), Communication (expressive), and Reading capacity). Additionally, their language profile was evaluated with the Linguistic Assessment for Mapped Provision test (LAMP) ([60]) whereas EFs were assessed with the Greek version of the Childhood Executive Functioning Inventory (CHEXI) ([82]) and the ADHD-IV Questionnaire ([26]).

## 2. Methods

### 2.1. Participants and Procedure

For the purpose of the present study, 71 preschool children were evaluated by their 20 teachers. Regarding the children’s sample, 41 (57.7%) were boys and 30 (42.3%) were girls. Specifically, of the total sample of participants, 60 children (84.5%) (between 4 and 6 years) attended kindergarten schools, and 11 children (15.5%) attended state or private childcare facilities. Regarding the gender of the teachers, all were women. Furthermore, the mean of teachers’ years of education experience was 10.83 years (*SD* = 5.58). Regarding the question, how well teachers feel that they know the student for whom they completed the questionnaires, for 39 students (54.9%) teachers answered that they felt they know the child well enough, for 14 students (19.7%) said that they feel they know the child very well, and for 18 students (25.4%) said they do not feel like they know the child very well. As regards to the time that the teacher knew the student for whom she completed the assessment instruments, 54.9% said they know the children well enough, 19.7% said that they know the children very well, and 25.4% answered that they do not know the children well. Specifically, for 20 children (28.2%), teachers declared that they knew the child from 1 to 3 months, for 8 children (11.3%) they answered that they knew the child from 4 up to 6 months, for 23 children (32.4%) they knew the child from 7 up to 12 months, and for 20 children (28.2%) they knew the child for more than a year.

Out of the 71 children, just one (1.4%) did not speak Greek as his/her first language, but for the other 70 (98.6%) Greek was their mother tongue. Teachers reported that 8 children had issues with written speech, whereas 63 children had no problems at all with written production. In terms of the learning profile of participants, teachers reported that, of the 71 students, 5 children had speech and language difficulties, with no official diagnosis and 1 child had articulation difficulties. Participants were recruited from kindergarten schools and/or childcare facilities of different geographical regions in Greece as the collection of the data was conducted (a voluntary task during the attendance of a “Cognitive Psychology” module) by students from the Department of Early Childhood Education of the University of Ioannina in Greece under the supervision of one of the authors. The exclusion criteria for every child’s participation were: (a) the child has not been diagnosed with any type of psychiatric or neurological disorder, (b) a diagnosis such as Attention Deficit/Hyperactivity Disorder (ADHD), or other neurodevelopmental diagnoses, (c) hearing impairments or any other sensory impairment, (d) not attending an inclusion class or special education services. Table 1 depicts all the criteria for participation according to the LAMP test.

### 2.2. Assessment Instruments

For the assessment of language ability, Greek preschool students were assessed by their preschool teachers with the Linguistic Assessment for Mapped Provision (LAMP) ([60]). EFs and WM were evaluated with the Greek version of the Childhood Executive Function Inventory (CHEXI) ([82]) and the ADHD-IV Questionnaire ([26]). Giftedness was assessed with the Greek version of the Gifted Rating Scales-Preschool/Kindergarten Form (GRS-P) ([62]) and creativity was evaluated with the Scales for Rating the Behavioral Characteristics of Superior Students (SRBCSS) ([69]).

#### 2.2.1. Linguistic Assessment for Mapped Provision (LAMP; [60])

The LAMP test ([60]) is an original assessment test which is intended to be used as a universal screening instrument for all primary school children. Its primary goal is linguistic assessment, which aims to support educators in accurately identifying students’ language and communication needs. The second goal is to assist educators in better focusing their instruction by utilizing the LAMP screening results as a reference for the types and severity of language needs of the students. Teachers must evaluate, through 41 proposals-questions, four linguistic areas (Expressive Language, Receptive Language, Behavior linked to Language and Social Communication). Specifically, 12 questions concern expressive language skills, such as articulation, phonology, and difficulty in producing sounds in words and sentences, 12 questions evaluate receptive language including understanding and processing incoming language, and 10 questions regard behavior related to language and communication skills. These behaviors included difficulties in starting and completing tasks and taking part in situations where talking was involved. Finally, 7 questions on social skills, such as the child’s difficulty in “maintaining non-verbal communication with others”. The purpose of the screening assessment test is to accurately identify children with deficits in speech, language, and communication abilities. The LAMP assessment test is based on a Likert-type scale with each sentence to be scored on a 4-point scale (Never = 0, Sometimes = 1, Frequently = 2, Constantly = 3). This screening test helps to establish the level of severity of the deficit that the child experiences. A low score indicates the child’s typical development in speech, language, and communication abilities; a high score indicates challenges in these areas and the need for further evaluation. The LAMP assessment scales were translated into Greek and were administered to Greek teachers by [44] ([44]).

#### 2.2.2. Childhood Executive Function Inventory (CHEXI; [82])

The Childhood Executive Function Inventory (CHEXI) ([82]) consists of 26 items. In this questionnaire, all the sentences/questions are associated with problems in EF, which can be perceived in children’s daily functions. More specifically, the 26 items of the CHEXI had initially been divided into four a priori scales based on Barkley’s ([6]) hybrid model: Working Memory (11 items), Planning (4 items), Inhibition (6 items), and Regulation (5 items). However, following the application of exploratory component analyses and the removal of items 25 and 26, [82] ([82]) offered a two-factor solution consisting of two distinct and easily comprehended factors. Given that planning is sometimes regarded as a more sophisticated WM function, the first element was interpreted as working memory (CHEXI-WM). The two scales measuring motivation regulation and inhibition were part of the second factor. The second scale was called inhibition (CHEXI-I) because these items taken together can be regarded as measuring both the cognitive and motivational components of inhibitory control. Teachers or parents can use a 5-point Likert scale to rate how well each line in this questionnaire characterizes the child: 1 represents completely not true, 2 represents not true, 3 represents mostly true, 4 represents true, and 5 represents absolutely true. A test of the factor structure of the Childhood Executive Function Inventory (CHEXI) in the Greek population has been implemented by [79] ([79]), who divided CHEXI into two scales: Inhibition Scale and Working Memory Scale. The internal consistency reliability of the Greek version of the CHEXI was evaluated with Cronbach’s alpha coefficient. The reliability of the Inhibition Scale was very good with a Cronbach’s *a* = 0.874. The reliability of the Working Memory subscale was excellent with a Cronbach’s *a* = 0.962.

#### 2.2.3. ADHD Rating Scale-IV (ADHD Rating Scale-IV; [26])

The scale is a norm-referenced checklist that measures the symptoms of Attention Deficit/Hyperactivity Disorder (ADHD) according to the diagnostic criteria of the Diagnostic and Statistical Manual of Mental Disorders (DSM-IV; [2]). The ADHD-Rating Scale-IV is standardized in a sample of the Greek population ([43]). The purpose of this scale is to provide clinicians with a means of gathering information regarding the frequency of certain behaviors from parents and teachers. The scale is completed independently by the parent or teacher, who reports the frequency of the symptoms over the past 6 months (or the beginning of the school year if the teacher has not known the student for 6 months) on a 4-point Likert scale (0 = Never, 1 = Very rarely, 2 = Sometimes, 3 = Very frequently). The ADHD-Rating Scale-IV is an 18-item questionnaire that takes approximately 15 min to complete. Two scales are distinguished from the ADHD-Rating Scale-IV: Inattention and Hyperactivity/Impulsivity. It consists of 9 questions about Attention Deficit and 9 questions about Hyperactivity/Impulsivity. Normative data are provided for ages 5 to 18. The reliability of the internal consistency of the Hyperactivity/Impulsivity scale, in the context of the present study, was excellent with a Cronbach’s *a* = 0.906. The reliability of the Inattention scale was also excellent with a Cronbach’s *a* = 0.907.

#### 2.2.4. The Gifted Rating Scales-Preschool/Kindergarten Form (GRS-P; [62])

The Gifted Rating Scales include a Preschool/Kindergarten Form (GRS-P; [62]) for ages 4.0 to 6.11 and are designed to be user-friendly. The GRS-P consists of five scales (namely: Intellectual, Academic, Creativity, Artistic, and Motivation) with 12 items each (60 items). Each item in every scale is rated by a teacher on a 9-point scale divided into three ranges: 1 to 3 = below average, 4 to 6 = average, and 7 to 9 = above average. This rating system allows the teacher to determine first whether the child is below average, average, or above average for each item compared to other students the same age and then to rate the student more specifically on a 3-point scale within the range. The scale consists of five scales: Intellectual Ability Scale. This scale measures the teacher’s ratings of a student’s verbal and/or nonverbal mental skills, capabilities, or intellectual competence. Academic Ability Scale: This scale measures the teacher’s ratings of the student’s skill in dealing with factual and/or school-related material. Creativity Scale: This scale measures the teacher’s ratings of the student’s ability to think, act, and/or produce unique, original, novel, or innovative thoughts or products. Artistic Talent Scale: This scale measures the teacher’s ratings of the student’s potential for, or evidence of, ability in drama, music, dance, drawing, singing, playing a musical instrument, and/or acting. Motivation Scale: This scale refers to the teacher’s perception of the student’s persistence, desire to succeed, tendency to enjoy challenging tasks, and ability to work well without encouragement. For the translation of the GRS-P Inventory in Greek, the International Test Commission (ITC) guidelines (www.intestcom.org) were followed. The back translation procedure was also followed to eliminate any inconsistencies that would disrupt the accuracy of the results. The Gifted Rating Scales-Preschool/Kindergarten Form (GRS-P) was checked for its psychometric properties in Greek preschool children ([79]). The internal consistency reliability of the scales of the Greek version of the GRS-P was evaluated with Cronbach’s alpha coefficient. Cronbach’s internal consistency of the three scales of the GRS-P was excellent and ranged between 0.979 and 0.981. The alpha internal consistency coefficient of the Intellectual Ability Scale was excellent *a* = 0.979, for the Academic Ability the indicator was *a* = 0.979, and for the Creativity Scale *a* = 0.981.

#### 2.2.5. Scales for Rating the Behavioral Characteristics of Superior Students (SRBCSS; [69])

The Scales for Rating the Behavioral Characteristics of Superior Students (SRBCSS) ([69]) were designed to evaluate the characteristics of the behavior of gifted students according to teachers’ estimation, based on the Three-Ring Model developed by [67] ([67]). In an attempt to offer a more objective assessment, the Scale for Rating Behavioral Characteristics of Superior Students (SRBCSS) is a screening test that teachers can utilize to guide their judgments during the identification process. Specifically, the Scale for Rating Behavioral Characteristics of Superior Students (SRBCSS) consists of 14 scales, which describe the abilities of students in the following areas: Learning Characteristics, Creativity Characteristics, Motivation Characteristics, Leadership Characteristics, Artistic Characteristics, Musical Characteristics, Dramatics Characteristics, Communication Characteristics (Precision), Communication Characteristics (Expressiveness), Planning Characteristics, Mathematics Characteristics, Reading Characteristics, Technology Characteristics, and Science Characteristics. Teachers rate students on the frequency of the behaviors displayed on a six-point Likert-type scale from ‘never’ to ‘always’. For the translation of the SRBCSS into the Greek language by Gitona, Tzalla, Foti, and Papantoniou, the International Test Commission (ITC) guidelines (www.intestcom.org) were followed. The back translation procedure was also followed to eliminate any inconsistencies that would disrupt the accuracy of the results. The SRBCSS was tested regarding its psychometric properties, using a sample of Greek teachers, by [84] ([84]). The reliability of internal consistency was excellent for the SRBCSS scales examined in this research. For the present study, we utilized the 5 scales from the Scales for Rating the Behavioral Characteristics of Superior Students (SRBCSS) ([69]). In specific, the scale of Learning Characteristics had an indicator of Cronbach’s *a* = 0.969. The index of the scale of Creativity Characteristics was *a* = 0.921, for the Communication Characteristics (Expressiveness) scale *a* = 0.957, and Reading Characteristics scale *a* = 0.925. Finally, the Communication Characteristics (Precision) scale had an index *a* = 0.957.

### 2.3. Procedure

Each educator was initially updated on the purpose of the current research and emphasized the anonymity of the answers. Each participant confirmed their consent to participate in the current study. A letter outlining the research aims, a demographic completion form, and the translated Greek versions of all the questionnaires, were given to each teacher by the researchers. The parents of the students also received a consent form. Each teacher received written and spoken information about the study, as well as the chance to ask questions. Every educator was given the option to select the location and time for filling out the questionnaires. Where appropriate, the researcher was present during the completion process to offer support or clarifications. To guarantee the validity of the outcome, educators were urged to provide authentic answers. Teachers were requested to complete the ratings on an individual basis, just based on their observations—not on deductions. All participants were advised that they might leave the evaluation process at any point, and there was no time limit on completing the scales. Participants were chosen from schools in various Greek regions. The study’s protocol followed the principles outlined in the Helsinki Declaration and was approved by the Scientific and Ethics Committee of the University of Ioannina (25847/01/06/2021).

### 2.4. Data Analysis

Although exploratory factor analysis (EFA) is useful in test construction, it does not provide an especially convincing test of the factorial structure of an inventory, as it does not permit the investigator to hypothesize and confirm which of a series of alternative plausible latent factor models best fits the data. Confirmatory factor analysis (CFA) is known to provide more detailed and sophisticated information regarding factor structure of the previously validated scales ([47]; [48]). Thus, both EFA and CFA were conducted in the current study to test the construct validity of the Greek version of the LAMP. Similarly, since path analysis evaluates more precisely the causal relationships among the examined variables ([10]; [16]), in the present study path analysis was preferred over multiple regression analysis. Structural equation models (both CFAs models and the path analysis model) were conducted in the statistical program EQS 6.1. ([10]).

The rest of the statistical analyses—which are a calculation of Cronbach’s alpha internal consistency reliability coefficients and Pearson’s *r* correlation coefficients examining the associations between the variables under analysis and the convergence of LAMP test ([60]) with respect to other instruments ([26]; [62]; [82]; [69])—were performed, with the use of IBM SPSS Statistics, version 25 and the statistical significance was set at 0.05.

## 3. Results

### 3.1. Test of the Factor Structure of Each of the Four Scales of the LAMP SCALES via Exploratory Factor Analyses Application

Initially, in the framework of the current study, the one-factor structure of each of the four LAMP scales ([60])—which were administered for the first time to Greek kindergarten teachers for the linguistic assessment of preschoolers—was examined by applying Exploratory Factor Analyses (EFA) for each scale individually. For the extraction of the factors, a principal component analysis with orthogonal Varimax rotation was used. Barlett’s sphericity control allowed for each of the four Kaiser–Mayer–Olkin measures to be utilized to assess the sample’s general suitability as well as the data’s suitability for the factor analysis. The original one-factor structure was validated for each of the four LAMP scales. The factors’ loading for all scales were found high.

#### 3.1.1. Test of the Factor Structure of the Expressive Language Scale of the LAMP

The Kaiser–Mayer–Olkin measure was applied to evaluate the total sample suitability, the value of which was K.M.O. = 0.903. Barlett’s sphericity control was statistically significant with *χ*^2^ = 561.253, df = 66, and *p* < 0.001. The analysis revealed one factor with an eigenvalue > 1.0. The eigenvalue of the factor was 6.885 and the percentage of the explained variance was 57.378%.

#### 3.1.2. Test of the Factor Structure of the Receptive Language Scale of the LAMP

The Kaiser–Mayer–Olkin measure was used to evaluate the sample suitability, which was K.M.O. = 0.907. For a further and more concise evaluation of the suitability of the data for factor analysis, Barlett Sphericity Test *χ*^2^ = 676.524, df = 66, and *p* < 0.001 were applied. The analysis of the data showed one factor with an eigenvalue > 1.0. The eigenvalue of the factor was 7.527 and the percentage of explained variance was 62.724%.

#### 3.1.3. Test of the Factor Structure of the Behavior Linked to Language Scale of LAMP

The Kaiser–Mayer–Olkin measure was used to assess the total sample suitability, the value of which was K.M.O. = 0.883. Barlett’s sphericity control was statistically significant with χ^2^ = 509.043, df = 45, and *p* < 0.001. The analysis revealed one factor with an eigenvalue > 1.0. The eigenvalue of the factor was 6.169 and the percentage of the explained variation was 61.693%.

#### 3.1.4. Test of the Factor Structure of the Social Communication Scale of the LAMP

The Kaiser–Mayer–Olkin measure was used to check the overall sample suitability, which was K.M.O. = 0.863. For a further and more complete examination of the suitability of the data for factor analysis, Barlett Sphericity Test *χ*^2^ = 468.339, df = 21, and *p* < 0.001 were performed. Data analysis revealed one factor with an eigenvalue > 1.0. The eigenvalue of the factor was 5.230 and the percentage of explained variation was 74.717%.

### 3.2. Test of the Factor Structure of Each of the Four Scales of the LAMP Test via Confirmatory Factor Analysis Application

Furthermore, to verify and evaluate the one-factor structure—that has been revealed via EFA—of each of the four scales of the LAMP assessment test for the Greek preschoolers sample, a set of four confirmatory factor analyses was conducted for the data collected from the proposals-questions for each of the four linguistic areas/LAMP scales [(1) Expressive Language, (2) Receptive Language, (3) Behavior linked to Language and, (4) Social Communication] on a four-point scale. Using the Maximum Likelihood estimation approach, each CFA was implemented in the statistical program EQS 6.1 ([10]) on a covariance matrix of the items that constitute each of the four LAMP scales, respectively. A non-statistical significance of the *χ*^2^ test indicates that the implied theoretical model significantly reproduces the sample variance–covariance relationships in the matrix ([48]). As this test is sensitive to sample size, model fit was also evaluated by using the root mean squared error of approximation (RMSEA). The RMSEA tests how well the model would fit the population covariance matrix. Specifically, a rule of thumb is that RMSEA ≤ 0.05 indicates close approximate fit, and values between 0.05 and 0.08 suggest reasonable error of approximation. Models with RMSEA = 0.10 (or RMSEA > 0.10) should be rejected ([48]; [16]). The Comparative Fit Index (CFI) which is one of the most popular incremental fit indices ([55]) assesses the relative improvement in the fit of the researcher’s model compared with a baseline model was also used. The CFI indicates a good model fit for values in the range between 0.95 and 1.00, whereas values in the range between 0.90 and 0.95 signify an acceptable fit ([12]). Additionally, the standardized root mean squared residual (SRMR) was used to evaluate the model fit. The mean absolute correlation residual, or the overall difference between the measured and predicted correlations, is measured by the SRMR. A favorable SRMR value is smaller than 0.08 ([10]; [9]; [15]).

The set of the initial four CFA models, that were conducted to test the one-factor structure of each of the four LAMP scales, produced the following indices:(1)Expressive Language Scale, *χ*^2^ (54, *N* = 71) = 119.41, *p* < 0.001, CFI = 0.88, SRMR = 0.07, RMSEA = 0.13 (CI 90% 0.10–0.16).(2)Receptive Language Scale, *χ*^2^ (54, *N* = 71) = 134.46, *p* < 0.001, CFI = 0.88, SRMR = 0.06, RMSEA = 0.15 (CI 90% 0.11–0.17).(3)Behavior Linked to Language Scale, *χ*^2^ (35, *N* = 71) = 97.37, *p* < 0.001, CFI = 0.87, SRMR = 0.07, RMSEA = 0.16 (CI 90% 0.12–0.20).(4)Social Communication Scale, *χ*^2^ (14, *N* = 71) = 74.99, *p* < 0.001, CFI = 0.87, SRMR = 0.06, RMSEA = 0.25 (CI 90% 0.19–0.30) ([9]; [47]; [48]).

Although all parameters of the aforementioned models were found to be statistically significant (*p* < 0.05) and the standardized root mean squared residual (SRMR) values were below 0.08, also indicating a good fit for the models tested, the chi-square goodness-of-fit test was statistically significant for all the initial models resulting in a rejection of the null hypothesis of good fit. In addition, the comparative fit index (CFI) values fell even more down from the lowest boundary of the marginal range of 0.90–0.95 and were indicative of poor model fit ([41]). Finally, the root mean squared error of approximation (RMSEA) values were above 0.13, indicating also a poor fit for the models tested. For these reasons, we proceeded with the identification of the areas of the initial models that contributed the most to the misfit. A residual analysis was conducted, and the Wald test was performed. Different models were tested, and the modifications indicated by the aforementioned tests were included in the model being tested each time. The modifications improved the fit of the final models on all indices and are as follows: (1)Expressive Language Scale: *χ*^2^ (48, *N* = 71) = 47.39, *p* = 0.498, CFI = 1.00, SRMR = 0.05, RMSEA = 0.00 (CI 90% 0.00–0.08). Standardized loadings of the measured variables on the latent variable (factor) ranged from 0.47 to 0.87, and R^2^ values for the latent variable (factor) ranged from 0.22 to 0.79.(2)Receptive Language Scale: *χ*^2^ (45, *N* = 71) = 58.37, *p* = 0.087, CFI = 0.98, SRMR = 0.04, RMSEA = 0.06 (CI 90% 0.00–0.11). Standardized loadings of the measured variables on the latent variable (factor) ranged from 0.57 to 0.88, and R^2^ values for the latent variable (factor) ranged from 0.33 to 0.77.(3)Behavior Linked to Language Scale: *χ*^2^ (28, *N* = 71) = 29.83, *p* = 0.371, CFI = 1.00, SRMR = 0.04, RMSEA = 0.03 (CI 90% 0.00–0.10). Standardized loadings of the measured variables on the latent variable (factor) ranged from 0.57 to 0.89, and R^2^ values for the latent variable (factor) ranged from 0.33 to 0.80.(4)Social Communication Scale: *χ*^2^ (7, *N* = 71) = 7.00, *p* = 0.429, CFI = 1.00, SRMR = 0.02, RMSEA = 0.00 (CI 90% 0.00–0.15) ([10]; [15]). Standardized loadings of the measured variables on the latent variable (factor) ranged from 0.63 to 0.92, and R^2^ values for the latent variable (factor) ranged from 0.40 to 0.85. In the second set of the CFA, all parameters of the final models were found to be statistically significant (*p* < 0.05) and the standardized root mean squared residual (SRMR) values were below 0.08, indicating also a good fit for the models tested. In addition, the comparative fit index (CFI) values were greater than 0.95. Specifically, the CFI indices in three of the four models were equal to 1.00 indicating an excellent model fit. The statistically non-significant chi-square goodness-of-fit test and the significant RMSEA values that were found in the final models were also indicative of their excellent fit.

### 3.3. Test of the Factor Structure of the Total LAMP Test via Confirmatory Factor Analysis Application

It should be noted that the aforementioned set of CFA at the item-level data—although they fully verified what was proposed by its constructor uni-factorial structure for each of the four scales of the Greek version of the LAMP—were limited as regards to the identification of a number of the LAMP organization’s underlying factors. Since the uni-factorial structure for each of the four scales of the Greek version of the LAMP was verified at the item-level data, and taking into account that a general factor (latent variable) is possible to account for most of the variance captured by the four LAMP ratings (measured variables), we applied confirmatory factor analysis (CFA), at the scale-level data, to all four LAMP scales of the Greek version in order for us to be able to identify the number of their organization’s underlying factors (latent variables). Since, taking into account the small sample size of the present study, the CFA was not run at the item-level but at the total scores for the aforementioned verified factor structure of each scale of the LAMP, the Expressive Language, the Receptive Language, the Behavior linked to Language, and the Social Communication, and scales were treated as measured (observed) variables in the CFA that was conducted on the scale-level data.

The CFA verified the one-factor structure—based on the aforementioned four measured variables—of the total LAMP for the sample of the present study [*χ*^2^ (0, *Ν* = 71) = 0.00, *p* = −1.00, NFI = 1.00]. The NNFI, CFI, and RMSEA were not computed (degrees of freedom = 0). The aforementioned model should be considered as just-identified ([16]), since it also contains two statistically significant correlations among independent variables (errors of measured variables). Standardized loadings of the four measured variables on the latent variable (general LAMP factor) ranged from 0.93 to 0.95, and R^2^ values for the general LAMP factor ranged from 0.86 to 0.91.

### 3.4. Test of the LAMP Internal Consistency Reliability

Cronbach’s alpha coefficients were also used to assess the LAMP internal consistency reliability. The Cronbach’s *α* internal consistency coefficients of all scales of the Greek version of the LAMP screen test, for the corresponding sample of the present study, ranged between 0.931 and 0.944 and were:(1)Expressive Language Scale, *α* = 0.931.(2)Receptive Language Scale, *α* = 0.944.(3)Behavior Linked to Language Scale, *α* = 0.928.(4)Social Communication Scale, *α* = 0.943.

The alpha internal consistency coefficients were outstanding for all scales of the LAMP Greek version. These findings closely match those of [60] ([60]).

### 3.5. Test of the Relationships Between Speech and Language Difficulties, EFs, and ADHD Characteristics

In the next step of statistical analysis to investigate the relationships between deficits in linguistic skills and EFs, as well as ADHD, the Pearson correlation was calculated between the LAMP test, CHEXI, and the ADHD-IV Rating Scale. A full correlation matrix among measures is provided in Table 2, which illustrates the statistically significant correlational relationship between children’s poor speech and linguistic skills performance with the ADHD and CHEXI rating scales. The application of correlation analysis revealed significant correlations between the LAMP test, CHEXI, and the ADHD-IV rating scales. Specifically, teachers relate speech and language deficiencies mainly to the inattention of ADHD as well as to working memory deficits of EF impairments. The LAMP test correlations with the ADHD-IV were all statistically significant. Specifically, the application of the Pearson analysis revealed a highly statistically significant correlation between the four scales of the LAMP test and the Inattention scale of the ADHD-IV rating scale (Expressive Language Scale *r* = 0.67, Receptive Language Scale *r* = 0.79, Behavior Linked to Language Scale *r* = 0.73, and Social Communication Scale *r* = 0.69). Additionally, the application of the Pearson analysis revealed a moderate-positive-statistically significant correlation between the LAMP test and the Hyperactivity/Impulsivity scale (Expressive Language Scale *r* = 0.46, Receptive Language Scale *r* = 0.41, Behavior Linked to Language Scale *r* = 0.43, and Social Communication Scale *r* = 0.45). Furthermore, the Pearson analysis showed statistically significant correlations between the four scales of the LAMP test and the two scales of the CHEXI Inventory. Specifically, the LAMP test is highly and positively correlated with the scale CHEXI-WM from the CHEXI Inventory (Expressive Language Scale *r* = 0.74, Receptive Language Scale *r* = 0.86, Behavior Linked to Language Scale *r* = 0.73, and Social Communication Scale *r* = 0.64) and moderately and positively correlated with scale CHEXI-I from the CHEXI Inventory (Expressive Language Scale *r* = 0.58, Receptive Language Scale *r* = 0.55, Behavior Linked to Language Scale *r* = 0.53, and Social Communication Scale *r* = 0.59). All significant results indicated that there was a considerable significant correlation between the LAMP test on the one hand, and EF measurements, on the other. Higher correlation values appear to be associated with WM and Inattention.

In the next step of statistical analyses, in order to evaluate more precisely the causal relationships, according to the above-mentioned variables, a path analysis was conducted. Considering that path analysis—a structural equation modeling (SEM) technique for analyzing structural models with observed variables—is adequate for examining relationships among multiple constructs measured using summated scales ([48]), and so we proceeded with this analysis. Specifically, to examine the relationships between EFs, ADHD, and speech and language deficits, a path analysis with manifest variables was computed. Because of the relatively small sample size, analysis was not run at the item level (observed variables). Instead, the covariance matrix was based on total scores (latent variables) for the LAMP, ADHD-IV, and CHEXI rating scales. The indicators of the LAMP test were defined as endogenous variables. The four ADHD and EF components were defined as exogenous variables. Path analysis was conducted in EQS 6.1 and was performed on a covariance matrix using the Maximum Likelihood estimation procedure ([10]). As shown in Figure 1, all indices were found to be positively associated. Specifically, the ADHD-IV Hyperactivity/Impulsivity scale from the ADHD-IV-Rating Scale was found to show a positive effect on the Expressive Language Scale from the LAMP test whereas the ADHD-IV Inattention subscale was found to positively affect the Receptive Language Scale, the Behavior linked to Language Scale, and the Social Communication Scale from the LAMP test. Additionally, the scale CHEXI-WM from CHEXI Inventory was found to have a positive effect on the Expressive Language Scale, the Receptive Language Scale, the Behavior linked to Language Scale, and the Social Communication Scale from the LAMP test. Finally, the CHEXI-Inhibition scale from the CHEXI Inventory was found to show a positive effect on the Expressive Language Scale and the Social Communication Scale from the LAMP test. The indices of this path model were excellent: *χ*^2^ (6, *N* = 71) = 6.03, *p* = 0.420, CFI = 1.00, SRMR = 0.02, RMSEA = 0.01 (CI 90% 0.00–0.15). The model is presented in Figure 1.

### 3.6. Test of the Relationships Between Speech and Language Deficits and Facets of Giftedness

In order to investigate the pattern of relationships between facets of giftedness and difficulties in speech, language, and communication abilities, the Pearson correlation coefficient between the LAMP test ([60]), GRS-Preschool/Kindergarten Form ([62]), and SRBCSS ([69]) rating scales were applied. As shown in Table 3, children’s deficits, as estimated by all the LAMP scales, were found to be negatively statistically correlated, with facets of giftedness. More specifically, the Intellectual Ability Scale shows negative statistical correlation with all four scales of the LAMP test (Expressive Language Scale *r* = −0.73, Receptive Language Scale *r* = −0.80, Behavior Linked to Language Scale *r* = −0.69, Social Communication Scale *r* = −0.66). Furthermore, the Academic Ability Scale is negatively correlated with the scales of the LAMP test (Expressive Language Scale *r* = −0.72, Receptive Language Scale *r* = −0.81, Behavior Linked to Language Scale *r* = −0.70, and Social Communication Scale *r* = −0.64). Finally, the Creativity Scale from the Gifted Rating Scales-Preschool/Kindergarten Form (GRS-P) also had a statistically significant negative correlation with the four scales of the LAMP test (Expressive Language Scale *r* = −0.70, Receptive Language Scale *r* = −0.78, Behavior Linked to Language Scale *r* = −0.70, and Social Communication Scale *r* = −0.62). The pattern of negative statistical correlation was also revealed with the five scales of the SRBCSS (Learning Characteristics, Creativity Characteristics, Communication Characteristics (precision), Communication Characteristics (expressiveness), and Reading Characteristics), and the LAMP Test. Specifically, the Learning Characteristics scale (SRBCSS) showed a statistically significant-negative correlation with the four scales of the LAMP test (Expressive Language Scale *r* = −0.69, Receptive Language Scale *r* = −0.76, Behavior Linked to Language Scale *r* = −0.70, and Social Communication Scale *r* = −0.66). Furthermore, the Creativity Characteristics scale also showed a negative statistical relationship with the LAMP test scales (Expressive Language Scale *r* = −0.52, Receptive Language Scale *r* = −0.65, Behavior Linked to Language Scale *r* = −0.62, and Social Communication Scale *r* = −0.51). A similar negative statistical pattern was revealed for the Communication Characteristics (precision) Scale (Expressive Language Scale *r* = −0.72, Receptive Language Scale *r* = −0.73, Behavior Linked to Language Scale *r* = −0.65, and Social Communication Scale *r* = −0.63), for the Communication Characteristics (expressiveness) (Expressive Language Scale *r* = −0.71, Receptive Language Scale *r* = −0.74, Behavior Linked to Language Scale *r* = −0.69, and Social Communication Scale *r* = −0.63) and for the Reading Characteristics Scale (Expressive Language Scale *r* = −0.61, Receptive Language Scale *r* = −0.63, Behavior Linked to Language Scale *r* = −0.58, and Social Communication Scale *r* = −0.52). Results revealed a slightly poorer correlation between reading proficiency, creativity, and linguistic deficiencies. Generally speaking, teachers appear to think that preschool children with speech and language difficulties are unlikely to be gifted. Table 3 depicts the correlations between facets of giftedness from the GRS-P, SRBCSS, and LAMP test.

### 3.7. Test of the Relations Between Giftedness with ADHD and EFs

Finally, to investigate the pattern of relationships between facets of giftedness and ADHD and EFs, the Pearson correlation coefficient between the GRS-Preschool/Kindergarten Form ([62]), ADHD-IV ([26]), and CHEXI Inventory ([82]) was applied. All facets of giftedness were found to be negatively statistically correlated with the ADHD-IV Scales (Inattention Scale and Hyperactivity/Impulsivity Scale) and CHEXI (Working Memory Scale and Inhibition Scale). Specifically, the application of the Pearson correlation analysis revealed moderate negative statistically significant correlations for the Inattention Scale and the Hyperactivity/Impulsivity Scale of ADHD-IV and for the Working Memory and the Inhibition scales of CHEXI with all the GRS-P Scales and the four scales of the SRBCSS. The Inattention and Hyperactivity/Impulsivity scales of the ADHD-IV and the Working Memory and the Inhibition scales of CHEXI evaluated the deficits in executive functions and for this reason, the correlations of its scales with the GRS-P and SRBCSS had negative indices. Consequently, teachers tend to believe that a student with ADHD characteristics does not appear to be associated with these specific facets of giftedness. Table 4 depicts the correlations between facets of giftedness from the GRS-P and SRBCSS with the ADHD-IV and CHEXI rating scales.

## 4. Discussion

The current study aimed to enlighten the multifactorial pattern of relationships of speech and language deficits with executive function (EF) deficits, Attention Deficit/Hyperactivity Disorder (ADHD) and with facets of giftedness, in Greek preschool children according to their teachers’ evaluations. An additional essential aspect in the present research analysis focused on the investigation of the pattern of relationships between EFs, ADHD, and giftedness in preschool-age children.

Regarding our first hypothesis, research outcomes revealed that according to teachers’ perceptions, speech and language deficits are positively associated with executive functions deficits (Hyperactivity/Impulsivity and inhibition) and ADHD characteristics, verifying Hypotheses 1a and 1b. Specifically, the correlation matrices showed that there is a positive statistical pattern of relationships shared by speech and language deficits and impairments in WM and inhibition in preschool-age children. These research data seem to be strengthened by a series of studies, emphasizing the fact that preschool children with speech and language deficits perform more poorly on WM assessment tests reflecting the deficit association ([12]; [23]; [61]; [74]). Furthermore, compared to typically developing children, preschool children with speech and language deficits show more clinically substantial EF and WM deficits, based on teacher estimations ([1]; [87]; [91]). In parallel, according to [86] ([86]), ratings of EF using the BRIEF-Preschool version (BRIEF-P; [35]) revealed that the parents of preschool children with speech and language deficits report significantly more EF deficits relative to the parents’ perceptions of typically developing children. Specifically, parents highlighted problems with inhibition, shifting, emotional control, WM, and planning which are all different facets of EF. In line with the above, our research findings highlighted that between the two facets of EFs (WM and inhibition), WM appeared to be highly significantly associated with preschool children with speech and language deficits. The present outcomes are consistent with other research studies regarding the association of WM and speech and language deficits ([74]; [72]; [81]; [53]). A common pattern of deficits within EFs appears to be in WM, affecting a high percentage of children with speech and language deficits ([65]; [81]). According to the research findings of [77] ([77]), preschool children with speech and language deficits register WM scores below the mean score—reflecting deficits in WM—when compared to typically developing children. A series of research studies indicated that WM challenges, encompassing phonological, auditory, and verbal memory, were the most often seen disturbances in preschool children with speech and language deficits. These were accompanied by impairments in attention, processing speed, inhibition, planning, cognitive flexibility, and internalized speech ([57]; [65]; [70]). On the other hand, regarding Hypothesis 1c, the relationships that were found between giftedness manifestations as judged by educators, and the speech and language deficits of preschool students revealed that these weaknesses are negatively correlated, confirming Hypothesis 1c. Research studies have showed that gifted preschool children are characterized by focused attention, early reading skills, high verbal ability, a well-developed sense of humor, curiosity, determination, and the ability to make abstract connections in the learning process ([22]). Furthermore, research studies show that the ability to read independently from a young age is a cognitive trait of gifted children. They can identify the letters from different signs or product logos without the assistance of an adult and subsequently use them to read new words because of their strong synthesis abilities. In addition, they consistently expand their vocabulary and make full use of word interpretation as early as possible ([18]; [22]).

Concerning the evaluation of the relationships between the Hyperactivity/Impulsivity Scale and Inattention Scale of the ADHD-IV Inventory, working memory, the Inhibition Scale of the CHEXI Inventory, and the four scales of the Linguistic Assessment for Mapped Provision (LAMP) test, the application of path analysis revealed that the ADHD-IV Hyperactivity Scale had a positive effect on the Expressive Language Scale whereas the ADHD-Inattention Scale was found to affect positively the three scales of the LAMP test (Receptive Language Scale, Behavior linked to Language Scale, and Social Communication Scale). Since hearing and listening are active processes that are closely related to language development, the effect of hyperactivity and inattention may be explained by difficulties paying prolonged and focused attention to the speech and sounds in the environment ([25]). The fact that ADHD is a disorder that impacts a child’s attention, thinking, learning process, and social interaction—all of which are critical for language development—may help to explain the notable delay in their total language age, and specifically in receptive language, and expressive and semantic language age. Also, impairments in WM which is highly related to language deficits can be a cause ([39]; [34]). Specifically, researchers claim that decreased WM, both verbal and spatial, are among the cognitive deficits purported to be characteristic of ADHD and is associated with Expressive and Receptive speech ([19]). In parallel, the above pattern of relationship is in line with a series of empirical data confirming that symptoms of inattention and hyperactivity typically co-occur with poor social communication skills ([8]) and low levels of Expressive and Receptive language ([7]) in children with Attention Deficit/Hyperactivity Disorder (ADHD). These overlapping symptoms represent aspects of hyperactive and inattentive behavior as well as the cognitive challenges that cross conventional diagnostic classifications like ADHD ([36]). Additionally, regarding the relationship between EFs, the LAMP test path analysis showed that the CHEXI-WM scale had a positive effect on the Expressive Language Scale and Social Communication Scale. From an early age, there may be a connection between language and EF problems. According to research analysis there is probably a reciprocal and intricate interaction between language deficits and EF deficits. A causal hypothesis by [13] ([13]) attempts to explain the connection between EF and language deficiencies. According to this hypothesis, language processing is impacted by EF. Specifically, WM deficiencies, in children with language deficits, may limit vocabulary acquisition by inhibiting the lexicon’s phonological representations. Further, inhibition deficits could underline lexical access deficits and deficient vocabulary learning in children with language impairments ([37]; [33]). Logically, of course, it is reasonable to assume that if children cannot retain phonological information long enough to form a phonological representation of a word due to WM impairments, or if their slower speed does not allow them to keep up with the processing of a full sentence, their language abilities should be hampered.

Regarding the third hypothesis, the evaluation of structural validity results revealed that the four scales of the LAMP test ([60]) retain the factorial structure proposed by their manufacturer. Furthermore, concerning the convergent and divergent validity of the LAMP assessment test, the results moved in the expected direction. Specifically, the moderate to high positive associations of the LAMP screening test with measurements of ADHD and EF showed the good convergent validity of the LAMP screening test as regarding ADHD-IV and CHEXI. Accordingly, the moderate to high negative correlations between the LAMP scales and the GRS-Preschool/Kindergarten scales of giftedness and SRBCSS measures revealed that the LAMP scales have good divergent validity with the scales estimating facets of giftedness. In conclusion, the complex nature of linguistic ability emphasizes the need for teachers to utilize new screening tests to accurately identify and evaluate linguistic skills, and to study their relationships or intercorrelations with other cognitive domains. In this context, the present study provides sufficient data to argue that the Linguistic Assessment for Mapped Provision (LAMP) can be a valid and reliable tool for assessing speech and language deficits in a Greek preschool population.

Regarding the second hypothesis, research analysis has revealed that different facets of giftedness are negatively associated with ADHD deficits and EF impairments, confirming Hypotheses 2a and 2b. More specifically, poor WM and inhibition were negatively correlated with different facets of giftedness, according to teachers’ estimations, whereas the hyperactivity symptoms seemed to be negatively correlated to a less important statistical degree with facets of giftedness. Although some findings of other studies show that some children with ADHD may be characterized as gifted ([1]), in the present study according to teachers’ estimations a negative statistical correlation has been found between giftedness and two facets of ADHD, and mainly with the dimension of attention deficit. Also, creativity, as a facet of giftedness, was not associated with the hyperactivity characteristics of ADHD, a finding that aligned with a previous research outcome regarding the characteristics of ADHD ([76]). Regarding the relationship of deficits in EF with giftedness, it was found that deficits in inhibition and WM are negatively associated with giftedness, confirming Hypothesis 2b, a finding that is in line with the research of [34] ([34]). It was also found that creativity, as a facet of giftedness, is not associated with inhibition. This finding is not in line with the findings of [19] ([19]), where inhibition was found to be associated with creativity. Despite the fact that gifted students with ADHD frequently have challenges in the classroom, there is evidence that they have a higher potential for creative achievement. In fact, the findings of the study by [29] ([29]) indicate that the combination of hyperactivity and inattention fosters creativity; nevertheless, our findings contradict this finding. Furthermore, the research has confirmed the relationship between deficits in EF and characteristics of ADHD. Poor WM and inhibition have also been found in previous studies in preschool children with ADHD ([29]; [46]; [49]; [64]; [83]). In addition, the research has revealed that the type of hyperactivity in ADHD is associated, according to teachers’ evaluations, with a high degree of inhibition, and to a lesser degree, with WM. Additionally, the high association of hyperactivity characteristics with deficits in inhibition could interpret the lack of relationship found between these two types of ADHD and creativity. One possible interpretation could be the fact that teachers are, according to their experience, able to comprehend the negative connection between dimensions of ADHD, inhibition, and WM deficits with facets of giftedness, but not with the facet of creativity. On the contrary, according to the finding of the present study, Greek kindergarten teachers seem to evaluate the creativity of their young students as a cognitive function that is independent from their students’ hyperactivity and/or deficits of inhibition.

Furthermore, the results highlighted that Greek preschool teachers tend to believe that there is a relationship between speech and language deficits and deficits in cognitive areas, such as EFs and attention in preschool children. The evidence unambiguously emphasizes how crucial educators are to the accurate identification process of gifted children from preschool age up to school age. In this sense, early identification will be made easier by preschool teachers’ awareness of specific characteristics of gifted children.

The implications of the current study are related to the need for appropriate educational opportunities that capitalize on these strengths of gifted children with speech and language deficits. The findings of this study also revealed teachers’ strong reliance on screening assessments to accurately identify linguistic and EF deficits. Educators would welcome a screening tool that takes less time to complete, especially when used in classrooms to screen large numbers of students for full gifted testing, early school admissions decisions, and determining appropriate grade placement and or acceleration decisions. Since having access to scientific identification tools is a start in the right direction for gifted education in Greece, the study’s practical implications include the availability of a reliable and valid tools for identifying giftedness and language deficits. The current study stresses the importance of the development of language skills and because it links the levels not only to deficits but also to various facets of giftedness, since it is quite common for research on high abilities not to focus especially on this stage of childhood. Resulting, the current research emphasizes on enlightening a pattern of intercorrelations between giftedness and other specific areas like ADHD characteristics, EF deficits, and speech and language deficits. [46] ([46]) noted that gifted children’s overreliance on strengths inadvertently obscures ADHD and EF impairments and may lead to diagnostic errors of omission. Accurate identification is difficult. Finding a pattern of relationship or a collection of traits and how they combine to generate a diagnosis is more important in differentiating between giftedness, ADHD, EF deficits, speech and language deficits, and the combination of all parameters, than it is in identifying symptoms. Despite the fact that gifted related behavioral traits and ADHD symptoms share certain similarities or intercorrelations, the reasons behind and scope of these relationships differ. It has been argued that this overlap leads to a misdiagnosis of some gifted children as coping with ADHD. Therefore, it is necessary for researchers and educators to pay attention to the adaptation challenges faced by gifted children. In addition, it is of vital importance to provide appropriate measures for accurate identification and diagnosis in order to design specific intervention programs. Teachers most commonly find themselves determining whether children are likely candidates for special education programs out of all the adults that comprise a child’s support system. Furthermore, teachers are not trained diagnosticians, nor are they expected to make diagnostic decisions. However, teachers frequently lack the specialized training needed to recognize possible distinctions between gifted characteristics, language impairments, ADHD symptoms, and EF deficits when they overlap. Future studies need to focus on how gifted traits, ADHD traits, and language deficits interact with other comorbidities, assess the responsiveness of children to different programs and interventions, and also to investigate how professional development affects teachers’ perceptions. In parallel, it is of vital importance to discuss that issues regarding the identification of gifted students have perplexed the field almost since its inception. The number of students, content, teaching strategies, and management of a gifted education program are all significantly impacted by how gifted individuals are identified.

### Limitations

At this point, it is important to discuss some limitations on the present study. The current data were collected with teachers’ rating scales from their observations during the school year, and teachers’ perceptions are likely to be biased. Despite the fact that numerous studies have been conducted to investigate teachers’ attitudes and perceptions regarding gifted children with ADHD characteristics, EF deficits, and language impairments, there is currently no comprehensive picture of this problem. The literature review reveals that cultural differences are crucial in the identification process, as well as in forming perceptions and influencing behaviors ([76]). Multicultural studies have revealed differences in teachers’ perceptions of the evaluations and education of gifted children in each nation ([83]). Furthermore, organizing and enhancing students’ learning processes, identifying their needs, and providing them with respectful instruction in accordance with their abilities and demands are all the responsibilities of teachers. Therefore, in order to improve instruction and meet the requirements of accurate identification of a gifted child, it is necessary to consider teachers’ possible biased perceptions in order to design effective interventions. In addition, the small number of participants, from specific geographical regions of Greece, makes the sample not representative but opportunistic. As this preliminary study is part of a broader experimental design concerning giftedness, speech and language deficits, and EF impairments in Greek preschool children, further research data are under evaluation. More studies with a larger sample of teachers and children from different geographical regions of Greece are needed in order to draw safer conclusions about the relationship of speech and language deficits with executive functions, Attention Deficit/Hyperactivity Disorder (ADHD), and facets of giftedness in preschool children in Greek educational settings. Additionally, another limitation of the study was the fact that there were no data concerning the socio-economic status or information regarding the educational status of the parents of participants that could influence the results of the study. Future research should also investigate socioeconomic issues, evaluate teachers’ perceptions regarding gifted children, their needs, and school preparedness to accommodate these populations.

## Figures and Tables

**Figure 1 behavsci-15-00136-f001:**
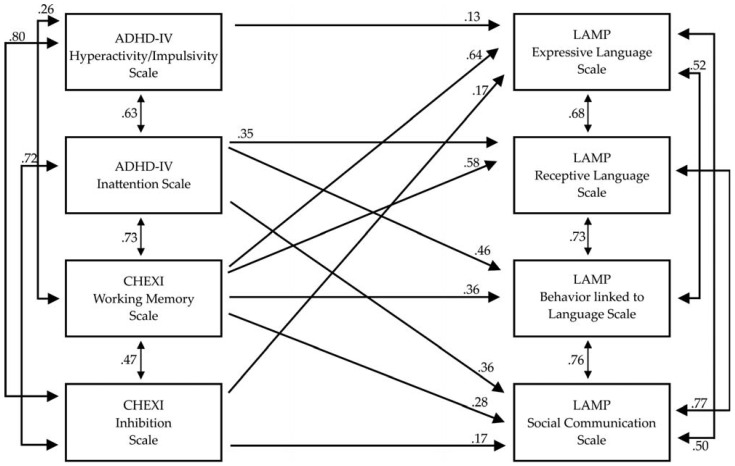
Path model displaying relationships among the components of ADHD, the deficits in executive functions, and the indicators of the Linguistic Assessment for Mapped Provision. Note: All loadings and parameters presented indicate statistically significant associations (*p* < 0.05) except for the effect of ADHD-IV Hyperactivity/Impulsivity Scale on the LAMP Expressive Language Scale (*p* = 0.17).

**Table 1 behavsci-15-00136-t001:** Information criteria according to the LAMP test demographic questions.

Participation Criteria	*f*	%
**First Language**		
Greek	70	98.6
Other	1	1.4
**Hearing Disorders**		
Hearing Disorders	0	0
Without Hearing Disorders	71	100
**Writing Difficulties Diagnosis**		
Writing Difficulties	8	11.3
Without Writing Difficulties	63	88.7
**Language Disorders**		
Language Disorders Diagnosis	0	0
Without Language Disorders	71	100
Inclusion Class Attendance	0	0
No Inclusion Class Attendance	71	100
**Type of Learning Difficulties**		
No specific type	65	91.5
Articulation difficulties	1	1.4
Speech difficulties	5	7

**Table 2 behavsci-15-00136-t002:** Pearson’s coefficients for the correlations between the ADHD-IV and CHEXI with the LAMP test.

	Expressive Language (LAMP)	Receptive Language (LAMP)	Behavior Linked to Language (LAMP)	Social Communication (LAMP)
Inattention-Scale (ADHD-IV)	0.673 **	0.798 **	0.734 **	0.691 **
Hyperactivity/Impulsivity Scale (ADHD-IV)	0.468 **	0.419 **	0.432 **	0.456 **
CHEXI-Working Memory Scale	0.744 **	0.862 **	0.732 **	0.648 **
CHEXI-Inhibition Scale	0.588 **	0.556 **	0.533 **	0.595 **

Note: ** Correlation is significant at the 0.01 level (2-tailed).

**Table 3 behavsci-15-00136-t003:** Correlations between facets of giftedness from the GRS-P and SRBCSS, and the LAMP test.

	Expressive Language (LAMP)	Receptive Language (LAMP)	Behavior Linked to Language (LAMP)	Social Communication (LAMP)
GRS-Preschool/Kindergarten Form				
Intellectual Ability	−0.734 **	−0.807 **	−0.696 **	−0.661 **
Academic Ability	−0.721 **	−0.818 **	−0.704 **	−0.649 **
Creativity	−0.707 **	−0.786 **	−0.708 **	−0.621 **
Scales for Rating the Behavioral Characteristics of Superior Students (SRBCSS)				
Learning Characteristics	−0.696 **	−0.769 **	−0.706 **	−0.668 **
Creativity Characteristics	−0.524 **	−0.654 **	−0.627 **	−0.517 **
Communication Characteristics (precision)	−0.724 **	−0.731 **	−0.657 **	−0.632 **
Communication Characteristics (Expressiveness)	−0.717 **	−0.744 **	−0.694 **	−0.634 **
Reading Characteristics	−0.617 **	−0.630 **	−0.589 **	−0.529 **

Note: ** Correlation is significant at the 0.01 level (2-tailed).

**Table 4 behavsci-15-00136-t004:** Correlations between facets of giftedness from the GRS-P, SRBCSS, ADHD-IV, and CHEXI Inventory.

	Inattention Scale (ADHD-IV)	Hyperactivity/Impulsivity (ADHD-IV)	CHEXI- Working Memory	CHEXI-Inhibition
GRS-Preschool/Kindergarten Form				
Intellectual Ability	−0.693 **	−0.251 *	−0.832 **	−0.470 **
Academic Ability	−0.722 **	−0.302 *	−0.834 **	−0.465 **
Creativity	−0.661 **	−0.190	−0.785 **	−0.398 **
Scales for Rating the Behavioral Characteristics of Superior Students (SRBCSS)				
Learning Characteristics	−0.648 **	−0.238 *	−0.762 **	−0.448 **
Creativity Characteristics	−0.491 **	−0.093	−0.608 **	−0.218
Communication Characteristics (precision)	−0.613 **	−0.284 *	−0.698 **	−0.507 **
Communication Characteristics (Expressiveness)	−0.574 **	−0.249 *	−0.710 **	−0.371 **
Reading Characteristics	−0.618 **	−0.418 **	−0.589 **	−0.576 **

Note: * Correlation is significant at the 0.05 level (2-tailed). ** Correlation is significant at the 0.01 level (2-tailed).

## Data Availability

The original contributions presented in this study are included in the article. Further inquiries can be directed to the corresponding author.

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
