# Peer review of "Evaluating the Pattern of Relationships of Speech and Language Deficits with Executive Functions, Attention Deficit/Hyperactivity Disorder (ADHD), and Facets of Giftedness in Greek Preschool Children. A Preliminary Analysis"

_behavsci, 2025, doi:10.3390/bs15020136_

Round 1

Reviewer 1 Report

Comments and Suggestions for Authors

see the attached document

Author Response

Introduction

Comment 1: Organize the information more logically. Currently, the text jumps between general language development, specific disorders, and the study's aims. Structure the introduction with a more logical progression, moving from broad concepts to specific research questions. You might organize it around these themes: 1) Importance of early language development 2) Prevalence and impact of SLDs 3) Relationship between SLDs and EFs/ADHD 4) The role of giftedness 5) This study's contribution.  Clearly articulate the gap in the current literature that this study addresses. Highlight what is novel about the study's approach or focus within the existing body of knowledge and explicitly state how this study fills that gap. What specific questions are unanswered by prior research? What is the unique contribution of this study to the field?

Answer to Comment 1: Thank you very much for your comments. In an attempt to clarify and organize the content according to your comments we added information regarding the importance of language development language development [see lines 47-58, 59-66, 68-69). We proceeded with information regarding the prevalence of SLD (see lines 74-80, 83-87). Regarding the relationship of SLD, EF and ADHD and the role of giftedness, as well as the research gap new information was added (see lines 88-106 and 147-154, 182-186, 189-201). You can track all the changes as they are highlighted in yellow.

Comment 2: Rephrase the hypotheses to clearly state the expected relationships between the variables using precise statistical terms (e.g., positive correlation, negative correlation). Number your hypotheses appropriately for clear referencing in the results section. Emphasize the real-world implications of the research. Discuss how understanding the interplay of SLD, EF, ADHD, and giftedness can benefit early childhood intervention programs and educational practices.

Answer to Comment 2: Thank you very much for your comment. We used precise statistical terms. The hypothesis are presented in a sequenced order (Hypothesis 1a , 1b, 1c etc.). You can track all the changes as they are highlighted in yellow.

Comment 3: The introduction uses both "speech and language disorders" and "Speech and Language Difficulties." Choose one consistent term and use it throughout.

Answer to Comment 3: Thank you very much for your comment. We used the term Deficits   according to the manufacturer of the LAMP test (Nash, 2013).

Methods

Comment 4: Detail the recruitment strategy, including the process for selecting schools and teachers. Specify the inclusion/exclusion criteria for participants (e.g., age range, diagnostic criteria for SLDs, presence of other developmental disorders). Address why a convenience sample was used rather than a random sample. Present a more detailed demographic breakdown of the sample. Consider factors like age (specific ranges and mean/standard deviation), socioeconomic status (if possible), and any relevant family history information. Explain why the specific sampling method (convenience sampling) was used and what implications this has on generalizability.

Answer to Comment 4: Thank you very much for your comment.  We added information regarding demographic characteristics of the sample (see lines 259-273), as well as Table 1 regarding exclusion and inclusion criteria according to the four demographic questions of the LAMP test (Nash, 2013).  (see line 291-Table 1). Demographic data that we didn’t evaluate we added as limitations of the study (see lines 958-963). We added explanation regarding the sample (see lines 279-289). We also add information regarding the

Comment 4: Detail the procedures for handling missing data, outliers, and ensuring data integrity. Specify the statistical software used for the analysis

Answer to Comment 4: Thank you very much for your help.  We added information regarding the statistical analysis, the type of the implied statistical methods and the etiology of their application. (see lines 444-459).

Results

Comment 5: The results could be better organised, this is a suggestion: 1. Factorial Validity of the LAMP: (present key findings from the CFA, focusing on model fit and factor loadings); 2 Internal Consistency Reliability: (table summarizing Cronbach’s alpha for all measures); 3 Relationships between SLD and ADHD/EFs: (results organized by hypothesis; correlations, effect sizes, p-values); 4 Relationships between SLD and Giftedness: (results organized by hypothesis; correlations, effect sizes, p-values); 5 Convergent Validity of LAMP: (correlations between LAMP and other measures)

Answer to Comment 5: Thank you very much for your comment. It really helped us to clarify our results. We have conducted CFA, and we present all the factors loading (see lines 497-592). Also, we have incorporated a Path analysis model for the accurate presentation of causal relationships. We have also add the figure of all factor loading according to the model (see lines 636-660).  We have also add a sub section entitled: Test of the relations of between giftedness with ADHD and EFs (see lines 735 -753).

Discussion-Conclusion

Comment 6:

1)Frame the interpretation of findings cautiously, focusing on associations and correlations rather than making causal claims. Thoroughly discuss the implications of the study's limitations, especially the reliance on teacher reports, on the interpretation of the results.

  1. Compare the findings to those of similar studies, highlighting both similarities and differences. Provide a detailed discussion of the theoretical implications of your findings.
  2. Expand the limitations section to thoroughly discuss the potential influence of teacher bias, discuss how this bias might have influenced the results, particularly regarding the assessment of giftedness. Consider citing relevant literature on teacher bias in similar assessments. Acknowledge the limitations of the small sample size, discussing its impact on the study’s generalizability and the possibility of type II error. Explain why the cross-sectional design prevents conclusions about causality and the directionality of effects.
  3. Provide specific and actionable suggestions for future research that could address the limitations of the current study (e.g., larger and more diverse samples, longitudinal designs, use of objective measures).

Answer to Comment 6: Thank you for your comment. We have added new research studies and in order to interpret our findings by adding also path analysis interpretation (see lines 802-839, 857-890). We also added information regarding the direction for future research and the implication of the current study regarding intervention programs, as well as teachers bias (see lines 910-937, 940-953). We clarified our limitations (see lines 918-930).

Introduction

Comment 1: Organize the information more logically. Currently, the text jumps between general language development, specific disorders, and the study's aims. Structure the introduction with a more logical progression, moving from broad concepts to specific research questions. You might organize it around these themes: 1) Importance of early language development 2) Prevalence and impact of SLDs 3) Relationship between SLDs and EFs/ADHD 4) The role of giftedness 5) This study's contribution.  Clearly articulate the gap in the current literature that this study addresses. Highlight what is novel about the study's approach or focus within the existing body of knowledge and explicitly state how this study fills that gap. What specific questions are unanswered by prior research? What is the unique contribution of this study to the field?

Answer to Comment 1: Thank you very much for your comments. In an attempt to clarify and organize the content according to your comments we added information regarding the importance of language development language development [see lines 47-58, 59-66, 68-69). We proceeded with information regarding the prevalence of SLD (see lines 74-80, 83-87). Regarding the relationship of SLD, EF and ADHD and the role of giftedness, as well as the research gap new information was added (see lines 88-106 and 147-154, 182-186, 189-201). You can track all the changes as they are highlighted in yellow.

Comment 2: Rephrase the hypotheses to clearly state the expected relationships between the variables using precise statistical terms (e.g., positive correlation, negative correlation). Number your hypotheses appropriately for clear referencing in the results section. Emphasize the real-world implications of the research. Discuss how understanding the interplay of SLD, EF, ADHD, and giftedness can benefit early childhood intervention programs and educational practices.

Answer to Comment 2: Thank you very much for your comment. We used precise statistical terms. The hypothesis are presented in a sequenced order (Hypothesis 1a , 1b, 1c etc.). You can track all the changes as they are highlighted in yellow.

Comment 3: The introduction uses both "speech and language disorders" and "Speech and Language Difficulties." Choose one consistent term and use it throughout.

Answer to Comment 3: Thank you very much for your comment. We used the term Deficits   according to the manufacturer of the LAMP test (Nash, 2013).

Methods

Comment 4: Detail the recruitment strategy, including the process for selecting schools and teachers. Specify the inclusion/exclusion criteria for participants (e.g., age range, diagnostic criteria for SLDs, presence of other developmental disorders). Address why a convenience sample was used rather than a random sample. Present a more detailed demographic breakdown of the sample. Consider factors like age (specific ranges and mean/standard deviation), socioeconomic status (if possible), and any relevant family history information. Explain why the specific sampling method (convenience sampling) was used and what implications this has on generalizability.

Answer to Comment 4: Thank you very much for your comment.  We added information regarding demographic characteristics of the sample (see lines 259-273), as well as Table 1 regarding exclusion and inclusion criteria according to the four demographic questions of the LAMP test (Nash, 2013).  (see line 291-Table 1). Demographic data that we didn’t evaluate we added as limitations of the study (see lines 958-963). We added explanation regarding the sample (see lines 279-289). We also add information regarding the

Comment 4: Detail the procedures for handling missing data, outliers, and ensuring data integrity. Specify the statistical software used for the analysis

Answer to Comment 4: Thank you very much for your help.  We added information regarding the statistical analysis, the type of the implied statistical methods and the etiology of their application. (see lines 444-459).

Results

Comment 5: The results could be better organised, this is a suggestion: 1. Factorial Validity of the LAMP: (present key findings from the CFA, focusing on model fit and factor loadings); 2 Internal Consistency Reliability: (table summarizing Cronbach’s alpha for all measures); 3 Relationships between SLD and ADHD/EFs: (results organized by hypothesis; correlations, effect sizes, p-values); 4 Relationships between SLD and Giftedness: (results organized by hypothesis; correlations, effect sizes, p-values); 5 Convergent Validity of LAMP: (correlations between LAMP and other measures)

Answer to Comment 5: Thank you very much for your comment. It really helped us to clarify our results. We have conducted CFA, and we present all the factors loading (see lines 497-592). Also, we have incorporated a Path analysis model for the accurate presentation of causal relationships. We have also add the figure of all factor loading according to the model (see lines 636-660).  We have also add a sub section entitled: Test of the relations of between giftedness with ADHD and EFs (see lines 735 -753).

Discussion-Conclusion

Comment 6:

1)Frame the interpretation of findings cautiously, focusing on associations and correlations rather than making causal claims. Thoroughly discuss the implications of the study's limitations, especially the reliance on teacher reports, on the interpretation of the results.

  1. Compare the findings to those of similar studies, highlighting both similarities and differences. Provide a detailed discussion of the theoretical implications of your findings.
  2. Expand the limitations section to thoroughly discuss the potential influence of teacher bias, discuss how this bias might have influenced the results, particularly regarding the assessment of giftedness. Consider citing relevant literature on teacher bias in similar assessments. Acknowledge the limitations of the small sample size, discussing its impact on the study’s generalizability and the possibility of type II error. Explain why the cross-sectional design prevents conclusions about causality and the directionality of effects.
  3. Provide specific and actionable suggestions for future research that could address the limitations of the current study (e.g., larger and more diverse samples, longitudinal designs, use of objective measures).

Answer to Comment 6: Thank you for your comment. We have added new research studies and in order to interpret our findings by adding also path analysis interpretation (see lines 802-839, 857-890). We also added information regarding the direction for future research and the implication of the current study regarding intervention programs, as well as teachers bias (see lines 910-937, 940-953). We clarified our limitations (see lines 918-930).

Introduction

Comment 1: Organize the information more logically. Currently, the text jumps between general language development, specific disorders, and the study's aims. Structure the introduction with a more logical progression, moving from broad concepts to specific research questions. You might organize it around these themes: 1) Importance of early language development 2) Prevalence and impact of SLDs 3) Relationship between SLDs and EFs/ADHD 4) The role of giftedness 5) This study's contribution.  Clearly articulate the gap in the current literature that this study addresses. Highlight what is novel about the study's approach or focus within the existing body of knowledge and explicitly state how this study fills that gap. What specific questions are unanswered by prior research? What is the unique contribution of this study to the field?

Answer to Comment 1: Thank you very much for your comments. In an attempt to clarify and organize the content according to your comments we added information regarding the importance of language development language development [see lines 47-58, 59-66, 68-69). We proceeded with information regarding the prevalence of SLD (see lines 74-80, 83-87). Regarding the relationship of SLD, EF and ADHD and the role of giftedness, as well as the research gap new information was added (see lines 88-106 and 147-154, 182-186, 189-201). You can track all the changes as they are highlighted in yellow.

Comment 2: Rephrase the hypotheses to clearly state the expected relationships between the variables using precise statistical terms (e.g., positive correlation, negative correlation). Number your hypotheses appropriately for clear referencing in the results section. Emphasize the real-world implications of the research. Discuss how understanding the interplay of SLD, EF, ADHD, and giftedness can benefit early childhood intervention programs and educational practices.

Answer to Comment 2: Thank you very much for your comment. We used precise statistical terms. The hypothesis are presented in a sequenced order (Hypothesis 1a , 1b, 1c etc.). You can track all the changes as they are highlighted in yellow.

Comment 3: The introduction uses both "speech and language disorders" and "Speech and Language Difficulties." Choose one consistent term and use it throughout.

Answer to Comment 3: Thank you very much for your comment. We used the term Deficits   according to the manufacturer of the LAMP test (Nash, 2013).

Methods

Comment 4: Detail the recruitment strategy, including the process for selecting schools and teachers. Specify the inclusion/exclusion criteria for participants (e.g., age range, diagnostic criteria for SLDs, presence of other developmental disorders). Address why a convenience sample was used rather than a random sample. Present a more detailed demographic breakdown of the sample. Consider factors like age (specific ranges and mean/standard deviation), socioeconomic status (if possible), and any relevant family history information. Explain why the specific sampling method (convenience sampling) was used and what implications this has on generalizability.

Answer to Comment 4: Thank you very much for your comment.  We added information regarding demographic characteristics of the sample (see lines 259-273), as well as Table 1 regarding exclusion and inclusion criteria according to the four demographic questions of the LAMP test (Nash, 2013).  (see line 291-Table 1). Demographic data that we didn’t evaluate we added as limitations of the study (see lines 958-963). We added explanation regarding the sample (see lines 279-289). We also add information regarding the

Comment 4: Detail the procedures for handling missing data, outliers, and ensuring data integrity. Specify the statistical software used for the analysis

Answer to Comment 4: Thank you very much for your help.  We added information regarding the statistical analysis, the type of the implied statistical methods and the etiology of their application. (see lines 444-459).

Results

Comment 5: The results could be better organised, this is a suggestion: 1. Factorial Validity of the LAMP: (present key findings from the CFA, focusing on model fit and factor loadings); 2 Internal Consistency Reliability: (table summarizing Cronbach’s alpha for all measures); 3 Relationships between SLD and ADHD/EFs: (results organized by hypothesis; correlations, effect sizes, p-values); 4 Relationships between SLD and Giftedness: (results organized by hypothesis; correlations, effect sizes, p-values); 5 Convergent Validity of LAMP: (correlations between LAMP and other measures)

Answer to Comment 5: Thank you very much for your comment. It really helped us to clarify our results. We have conducted CFA, and we present all the factors loading (see lines 497-592). Also, we have incorporated a Path analysis model for the accurate presentation of causal relationships. We have also add the figure of all factor loading according to the model (see lines 636-660).  We have also add a sub section entitled: Test of the relations of between giftedness with ADHD and EFs (see lines 735 -753).

Discussion-Conclusion

Comment 6:

1)Frame the interpretation of findings cautiously, focusing on associations and correlations rather than making causal claims. Thoroughly discuss the implications of the study's limitations, especially the reliance on teacher reports, on the interpretation of the results.

  1. Compare the findings to those of similar studies, highlighting both similarities and differences. Provide a detailed discussion of the theoretical implications of your findings.
  2. Expand the limitations section to thoroughly discuss the potential influence of teacher bias, discuss how this bias might have influenced the results, particularly regarding the assessment of giftedness. Consider citing relevant literature on teacher bias in similar assessments. Acknowledge the limitations of the small sample size, discussing its impact on the study’s generalizability and the possibility of type II error. Explain why the cross-sectional design prevents conclusions about causality and the directionality of effects.
  3. Provide specific and actionable suggestions for future research that could address the limitations of the current study (e.g., larger and more diverse samples, longitudinal designs, use of objective measures).

Answer to Comment 6: Thank you for your comment. We have added new research studies and in order to interpret our findings by adding also path analysis interpretation (see lines 802-839, 857-890). We also added information regarding the direction for future research and the implication of the current study regarding intervention programs, as well as teachers bias (see lines 910-937, 940-953). We clarified our limitations (see lines 918-930).

Reviewer 2 Report

Comments and Suggestions for Authors

REVIEW ARTICLE:

Evaluating the pattern of relationships of Speech and Language 2 Disorders with Executive functions, Attention Deficit / Hyper-3 activity Disorder (ADHD) and facets of giftedness in Greek 4 preschool children. A preliminary analysis

The research presented here aims, as a first objective (1) to provide information on the relationships between speech and language difficulties and executive function (EF) deficits, attention deficit hyperactivity disorder (ADHD) and as a second objective (2) to investigate the relationship of EF with ADHD, and the relationship between these two parameters with different facets of giftedness. All this in preschool children in Greece, mainly attending to the evaluations of their teachers.

To this end, several hypotheses linked to the research objectives are proposed. For Objective 1, it establishes three hypotheses:

·       H1a: Impaired language skills are positively associated with deficits in EFs (MT and inhibition).

·       H1b: Language difficulties are positively associated with ADHD characteristics.

·       H1c: Speech and language disorders are negatively associated with facets of giftedness.

For Objective 2, three other Hypotheses are established:

·       H2a: There are different facets of giftedness that are negatively associated with ADHD.

·       H2b: There are different facets of giftedness that are negatively associated with EF impairments.

·       H2c: ADHD symptoms in gifted preschool students are positively associated with EF.

On the other hand, with respect to the tests used, two other Hypotheses are proposed:

·       H3a: For the convergent validity of the LAMP test with the Greek translated version of the Child Executive Functioning Inventory (CHEXI) and the ADHD Inventory - IV it is expected to achieve a positive correlation between three measurements.

·       H3b: The Linguistic Assessment for Mapped Provision (LAMP) correlates negatively with all three subscales of the Gifted Rating Scale-Preschool Form-Kindergarten Form (GRS-P), reflecting the level of discriminant validity between two assessment tests.

The topic seems relevant to me, especially because it stresses the importance of the development of language skills and because it links the levels not only to deficits but also to various facets of giftedness, since it is quite common for research on high abilities not to focus especially on this stage of childhood. Therefore, I believe that any contribution that helps to determine links between giftedness and other specific areas of study is relevant and enlightening.

I consider that the variety and quality of the tests chosen for data collection and analysis show well-designed research and that the interpretation of the results is rigorous and adequate. The hypotheses put forward are well formulated and the method used to answer each one of them is argued.

Regarding the methodology, I would like to emphasize that I agree with the authors on the limitations they themselves point out, related to the sample, which is not very representative, and the bias in the teachers' interpretation of their observations of the children. I am confident in their proposals for the future in that the research is still open and more data collected are being evaluated and I hope that in the future they will be able to provide more evidenced conclusions from more representative samples.

The results they present are based on a detailed analysis after the implementation of each of the tests they have used, and in the discussion, they present their reflections linked to each of the hypotheses established.

The 75 references used are highly representative as they address all the areas covered by the research and in each of them they have started from authors and researchers who are truly reliable and of the highest level, who have important publications and who are fundamental references in these fields of research.

I would like to point out that I found a small typo regarding the numbering of the references: on line 144, the reference to Cordeiro et al (2011) is numbered as [37] and should be [39].

Author Response

Dear Reviewer thank you very much for your comments. We have highlightes the modifications we have conducted the the hypothesis section according to your comments. You can trach all the changes highlightes in yellow. We have allso conducted new statistical analysis like CFA and Path analysis to enhance our research findings. 

Round 2

Reviewer 1 Report

Comments and Suggestions for Authors

The revised manuscript shows significant improvement in addressing the previously raised concerns. The changes made directly address most of the recommendations. The improvements in the Introduction, Methods, and Results sections are particularly noteworthy, enhancing the clarity and rigor of the study. The discussion section is also improved.

In conclusion, the revised manuscript is now suitable for publication.